# Electroencephalographic and Epilepsy Findings in ZNF711 Variants: A Case Series of Two Siblings

**DOI:** 10.3390/neurolint17010014

**Published:** 2025-01-20

**Authors:** Michele Minerva, Lorenzo Perilli, Samanta Carbone, Margherita Maria Rossi, Federica Lotti, Luisa Lonoce, Maria Rosaria Curcio, Salvatore Grosso

**Affiliations:** Clinical Pediatrics, Department of Molecular Medicine and Development, University of Siena, Azienda Ospedaliero-Universitaria Senese, 53100 Siena, Italy

**Keywords:** epilepsy, ZNF711, X-linked intellectual disability

## Abstract

Background/Objectives: ZNF711(Zinc finger protein 711) encodes a zinc finger protein of currently undefined function, located on the X chromosome. Current knowledge includes a limited number of case reports where this gene has been exclusively associated with X-linked intellectual disability (XLID). As far as we are aware, we report the first cases of epilepsy associated with this particular variant. Our aim is to further delineate the phenotypic spectrum of ZNF711 gene pathogenic variants, adding clinical features to this rare condition, following a genotype-first approach. Case presentation: We describe the familiar case of two male siblings presenting with moderate intellectual disability (ID), language delay, and motor stereotypies. Additionally, they experienced generalized tonic–clonic seizures (GTCSs) and myoclonic seizures with interictal electroencephalographic abnormalities. Both children underwent various genetic testing and counselling, including an extended next-generation sequencing (NGS) panel, revealing a hemizygous c.657C > G pathogenic variant in the ZNF711 gene from maternal inheritance. Conclusions: This case expands the clinical range of ZNF711 variants by highlighting epilepsy as a potential comorbidity and suggesting other possible causal candidates for generalized epilepsy. Moreover, it emphasizes the need for further research into the phenotypic spectrum associated with this variant.

## 1. Introduction

Intellectual disability (ID) is defined by an impairment of cognitive and functional abilities with an onset during developmental period [1]. It has been estimated that ID affects approximately 1–3% of the general population, with a higher prevalence among male children compared to female ones [2].

ID can be due to both congenital or acquired causes such as genetic anomalies, congenital infections, prematurity, perinatal complications, hypothyroidism, psychosocial deprivation, and malnutrition [1]. Genetic causes of ID include minor genetic anomalies such as deletions, mutations or duplications of single or multiple genes or frank chromosomal anomalies, e.g., Down syndrome, fragile X syndrome (FXS), Prader Willi syndrome, etc. These genetic anomalies may occur de novo or be transmitted from parents with different inheritance patterns. Overall, genes involved in ID are frequently located on the X chromosome and thus responsible for XLID, which clearly affects males more than females.

Historically, X-linked causes of ID have been deeply investigated as the male prevalence of ID raised the suspicion of the involvement of X-linked genes in pedigrees of families affected by ID. Nevertheless, the identification of these XLID genes is challenging as mutations of the majority of genes located on the X chromosome may not be directly related to a pathological phenotype [2]. Therefore, it may be important to report further evidence of the phenotypic spectrum of patients with ID and XLID gene mutations.

Among the XLID genes apparently involved in ID, ZNF711 gene mutations have been described as rare causes of non-syndromic ID. To date, only a few families carrying a mutation of this gene have been described in terms of cognitive outcome, phenotypic features, and associated comorbidities [3,4,5].

ZNF711 is a gene located in the Xq21.1–q21.2 region, encoding for a zinc finger protein (ZNF) whose function is still unknown. However, this protein presents an amino-terminal potential domain followed by 12 consecutive Zn-C2H2 domains, which seem to be involved in transcriptional activation and sequence-specific DNA binding, respectively [3].

Therefore, it has been hypothesized that ZNF711, like other ZNFs, may have a role in regulating gene expression at the transcriptional and translational levels, with a consequent effect on neurological development. Furthermore, it has been shown that ZNFs are involved in the sonic hedgehog signaling (SHH) pathway, which has been directly associated with seizure genesis [6].

According to the evidence currently present in the literature, ZNF711 seems to be associated with a non-syndromic mild-to-moderate ID which is associated with occasional comorbidities such as autistic features and speech delay [3].

On the contrary, no cases of ZNF711 mutations have been associated with epilepsy. Hence, we describe the clinical and electroencephalogram (EEG) features of two siblings carrying a hemizygous c.657C > G pathogenic variant in the ZNF711 gene affected by a non-syndromic form of ID and seizures.

This evidence supports the idea of a clinical relevance of the ZNF711 variant encountered in our patients, with both ID and seizures.

To our knowledge, no other cases of ZNF711 pathogenic variants resulting in ID and seizures have been described.

Considering the paucity of information regarding this rare genetic condition, our aim with this familiar case series is to extend the clinical features of the phenotypic spectrum non-syndromic ZNF711-related ID.

## 2. Case Report

We report a familiar case series of two male siblings presenting with moderate ID, language delay, motor stereotypies, and generalized seizures. Both children carry a hemizygous c.657C > G pathogenic variant in the ZNF711 gene inherited from the maternal side of the family, and their parents have no previous history of epilepsy or neurodevelopmental delays.

Case 1

Male patient born at term via cesarean section due to a failure to progress during labor after an unremarkable pregnancy history.

His psychomotor developmental milestones were characterized by independent walking at 15 months and a mild language delay, with initial words at around 2 years of age. Due to this evidence, at the age of 2, a neuropsychiatric evaluation was performed, highlighting problems in emotional/behavioral self-control with scarce tolerance to frustration. Furthermore, the child showed motor clumsiness and stereotypies. Upon physical examination, at the age of 2, he showed facial features including macrocephaly, sunken eyes, a flat philtrum, small and spaced teeth, and flat feet.

Subsequently, at the age of 3, he started experiencing episodes of symmetric myoclonic jerks to all four limbs, associated with eye roll back and loss of consciousness. Furthermore, the child showed a severe language delay with dysarthria, as well as expressive language consisting of simple sentences.

In order to confirm the suspicion of myoclonic seizures, a prolonged video-electroencephalogram (video-EEG) recording was performed which demonstrated a correlation between phenotypic expression and epileptic abnormalities. Therefore, antiseizure therapy with valproic acid (VPA) 30 mg/kg was initiated, resulting in a seizure-free outcome.

Additionally, the child underwent brain magnetic resonance imaging (MRI), which was unremarkable, as well as urinary and blood laboratory tests which yielded normal results, thereby excluding structural or metabolic causes.

Cognitive evaluation with the Griffiths Mental Development Scales performed at 3 years and 9 months of age showed a moderate ID (General Quotient, GQ, 56).

At the last clinical evaluation during his periodical follow-up (16 years), he showed persistence of motor stereotypies, a regression of the behavioral issues, and seizure-free results since VPA was started.

Expressive language is actually characterized by the use of numerous words with persistent difficulties in the fluency of speech thus requiring school support.

EEGs and Video-EEG Monitoring: The first EEG recordings, at the age of 3, showed normal background activity and non-rapid eye movement (NREM) sleep elements. Interictal activity was characterized by slow spike-waves in the right posterior regions and occasional polyspike-waves in the left mid-frontal regions. Generalized EEG discharges were mainly observed during sleep. Intermittent photic stimulation and hyperventilation had not triggered epileptic discharges.

During the long-term video-EEG monitoring performed at the age of 3, at the onset of epilepsy, myoclonic seizures were recorded. Such epileptic manifestations are depicted in Figure 1; they are characterized by jerks in the upper limbs with an ictal EEG showing generalized slow spike-waves and electromyography (EMG) signals congruent with myoclonic seizures (Figure 1).

The latest EEG recording (16.3 years) showed persistence of normal background activity, with few generalized interictal discharges and without clinical events.

Case 2

Male patient born at term via elective cesarean delivery, due to previous cesarean sections, after an uneventful pregnancy.

His psychomotor developmental milestones were characterized by independent walking at 18 months and a language delay, with his first words after the age of two and scarce progressive lexical acquisition. A neuropsychiatric evaluation conducted at 2 years old reported autistic features, motor clumsiness, and motor stereotypies. On that occasion, the objective examination showed adequate auxological parameters and—like his sibling—his facial features included sunken eyes, flat philtrum, small/spaced teeth, and flat feet.

Seizure onset occurred at the age of 4, contrary to his sibling, with GTCS. Akin to his brother, his metabolic tests and brain MRI results were normal. The patient had been initially treated with VPA (30 mg/kg) in monotherapy, with persistence of the ictal events, therefore an add-on clobazam therapy (10 mg/day) was prompted, leading to a seizure-free state.

At the age of 4 years, the patient was tested with the Griffiths Mental Development Scales, with the result of a moderate ID (GQ 47).

During the last clinical evaluation at 12 years old, the patient presented persistence of motor stereotypies, expressive language characterized mainly by bisyllabic words, supported by communication through pointing, and seizure-free results from the age of four years with VPA and Clobazam.

EEG and Video-EEG Monitoring: The first EEG was performed at seizure onset and showed normal background activity and NREM sleep elements. Interictal activity showed slow spike-waves in the right posterior regions, with additional spike-waves and polyspike-waves in the anterior regions (Figure 2).

Generalized epileptiform discharges were more frequent during sleep. Intermittent fotic stimulation and hyperventilation did not trigger epileptiform abnormalities.

At 5 years of age, non-paroxysmal events, like hand flapping and hypnagogic jerks, were recorded during a video-EEG monitoring session. The latest EEG at 12 years old showed persistence of a normal background activity, with interictal spike-waves in the left anterior regions and without clinical events.

*Genetics:* In order to define the etiology of ID and epilepsy, we performed in both patients an array comparative genomic hybridization (a-CGH) testing, which was normal and FXS was excluded. Subsequently, the family underwent an NGS of a wide panel of genes associated with neurodevelopmental disorders and seizures, which revealed in both siblings a c.657 > C (p.Ile22Thr) pathogenic variant in the exon 1 of the ZNF711 gene (NM_021998.4, GRCh37/hg19, OMIM *314990) in a hemizygous state from maternal inheritance.

## 3. Discussion

To date, about 30 genes, including 8 with zinc finger genes on the X chromosome, have been associated with a non-syndromic ID [5]. Among them, the ZNF711 gene is located within the Xq21.1–q21.2 region and encodes a zinc finger protein with still unclear function. It has been hypothesized that ZNF11 interacts with demethylase HF8 in the promoter region of common target genes, affecting their methylation and thus their expression [7]. ZNF711 has been proven to be expressed in certain areas of the brain and neural tissues, thus variants of this gene have been related to sporadic forms of non-syndromic XLID [3,4,5].

XLID involves variants in genes on the X chromosome resulting in syndromic or non-syndromic complex diseases, with the latter typically presenting without distinctive dysmorphisms or neurological features [8].

Since the discovery in 1991 of variants in the FMR1 gene causing FXS, the most common form of XLID currently known, numerous other genes have been accounted, including ZNF711 [9].

The state of the art has described only a limited number of case reports, highlighting the heterogeneous phenotypic features of patients carrying pathogenic variants of the ZNF711 gene (Table 1 and Table 2). To the best of our knowledge, by conducting a narrative review of the literature, about 10 variants of ZNF711 have been reported to be associated with ID (Table 2).

Van der Werf et al. described 11 patients from two families with XLID caused by ZNF711 variants. All the male patients shared mild to moderate ID and speech delay, with some of them presenting additional autistic features, including flapping hands and avoidance of eye contact. Some minor nonspecific facial dysmorphisms such as a rather broad face and prominent forehead were also noted (Table 1) [3].

Another case series by Wang et al. included male patients with ZFN711 variants exhibiting mildly impaired cognitive function, associated with a slight delay in the achievement of motor and expressive language development. Autistic features were observed in approximately half of the cohort. Interestingly, peculiar physical features were observed in the majority of patients, including large head circumference, downslanting palpebral fissures, deep-set eyes, thin nose and alae nasi, prominent chin, upturned earlobes, and thin upper lip (Table 1) [5].

Similarly, Liang et al. described the case of a syndromic male patient with choroideremia, deafness, and ID. This patient carried a contiguous Xq21 gene deletion, including the POU3F4, CHM, and ZNF711 genes. The authors suggested that POU3F4 and CHM gene deletion could explain the hearing and visual loss while ZNF711 may be the putative gene for the ID and facial dysmorphisms (narrow forehead, dropping eyelids, and short palpebral fissures) presented by the patient (Table 1) [4].

Conversely, Tarpey et al. did not report any pathognomonic dysmorphic features in patients carrying the ZNF711 variants associated with XLID. They performed the sequencing of X chromosome coding exons in 208 families with ID. The screening discovered nine genes, including ZNF711, whose variants were associated with moderate ID without dysmorphic features [10].

Our patients presented minor facial dysmorphisms such as sunken eyes, flat philtrum, and small and spaced teeth, which have never been previously described. This may support the evidence that phenotypic dysmorphic features of patients carrying pathogenic ZNF711 gene variants are heterogeneous and thus could not be considered pathognomonic of a specific syndrome.

To the best of our knowledge, our siblings are the first two patients affected by a ZNF711-related ID in comorbidity with seizures.

Specifically, both the patients experienced generalized seizures. In patient 1, myoclonic seizures occurred, while patient 2 exhibited GTCS. Video-EEG recordings confirmed myoclonic seizures in patient 1, while both siblings displayed many non-epileptic stereotyped movements. In both cases, interictal EEG recordings showed normal background activity but revealed focal, sleep-activated, paroxysmal activity. In terms of antiseizure medications, VPA in monotherapy was found to be effective in controlling seizures in patient 1, whilst an association of VPA and Clobazam resulted in patient 2 being seizure-free.

There is paucity of information regarding the physiopathological role of ZNFs in seizures. However, several studies have demonstrated that these genes may have a role in the transcriptional and translational regulation of other genes involved in central nervous system development through the SHH signaling pathway [6,11,12]. Indeed, some proteins belonging to the same family of ZNF711 have been proven to have a relationship not only with the neuronal inflammation and oxidative stress seen in stroke and other diseases but also in the stem cell proliferation and neuronal differentiation associated with autism and schizophrenia [6].

It has been widely demonstrated that SHH is involved in epilepsy through an enhancement of extracellular glutamate levels which triggers neuronal excitation thus leading to epilepsy [12]. With ZNF as important modulators of SHH signaling pathways, we postulate the role of these genes in neurological development and epilepsy [11].

In our familial cohort, we present the first case series of patients with ID and epilepsy, likely caused by a pathogenic variant in the ZNF711 gene, expanding the known clinical spectrum of these variants. As no other genetic causes have been identified and given the scientific literature suggesting a possible causative role of the ZNF711 gene variant, we believe it is plausible that the clinical picture of ID and seizures of the two siblings may be related precisely to this variant found.

Therefore, our study suggests that zinc finger genes should be further investigated as a novel potential gene candidate in the wide spectrum of genes involved in genetically determined generalized epilepsy.

## 4. Conclusions

In conclusion, we described the first documented cases of ID associated with seizures in probands carrying a hemizygous c.657C > G pathogenic variant in the *ZNF711* gene. This case is limited by the anecdotal nature of the report. While additional studies are required to better understand the specific role of ZNF711 variants in the development of epilepsy, it expands the clinical spectrum of ZNF711 variants by highlighting epilepsy as a possible comorbidity. Furthermore, it adds other potential causal candidates for generalized epilepsy. Given the limited number of reported cases, further studies are required to broaden the understanding of the phenotypic spectrum and to explore genotype–phenotype correlations in ZNF711 variants.

## Figures and Tables

**Figure 1 neurolint-17-00014-f001:**
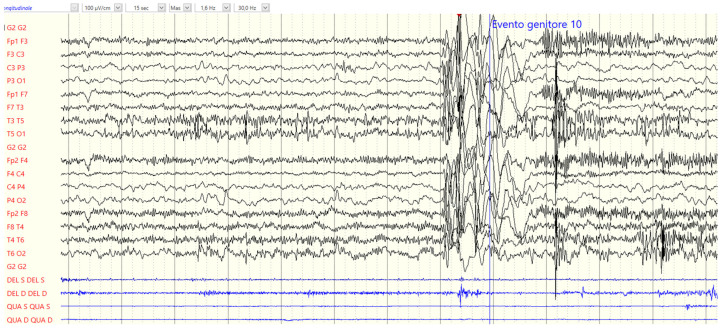
Video-EEG monitoring at the age of 3 years showing myoclonic seizures in patient 1.

**Figure 2 neurolint-17-00014-f002:**
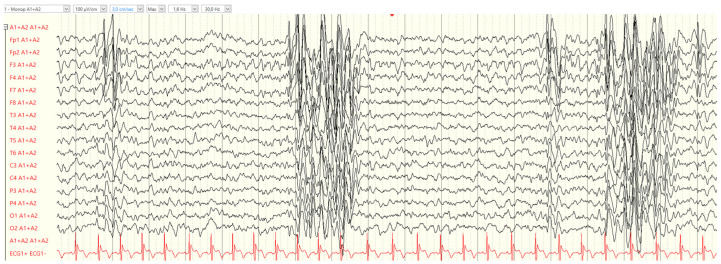
Interictal EEG at the age of 4 years showing spike-waves and polyspike-waves in the anterior regions and generalized abnormalities in patient 2.

**Table 1 neurolint-17-00014-t001:** Clinical findings in patients with ZNF711 variants. Abbreviations: U = unknown.

	van der Werf et al., 2019 [4]	Wang et al., 2022 [10]	Liang et al., 2017 [9]	This Paper
Male affected	11	20	1	2
ZNF711 variants	11/11	20/20	1/1	2/2
Other genes variants	0/11	0/20	Xq21 deletion	0/2
Mild ID	6/11	17/20	0/1	0/2
Moderate ID	5/11	1/20	1/1	2/2
Autistic features	2/11	8/20	U	2/2
Seizures	0/11	0/20	0/1	2/2
Speech delay	10/11	U	1/1	2/2
Motor delay	4/11	U	1/1	0/2
Craniofacial dysmorphisms	11/11	20/20	1/1	2/2
Others symptoms	U	U	choroideremia, deafness	U

**Table 2 neurolint-17-00014-t002:** ZNF711 variants.

van der Werf et al., 2019 [4]	Wang et al., 2022 [10]	Liang et al., 2017 [9]	Tarpey et al., 2009 [10]	This Paper
p.(Phe685Serfs*7)	p.Arg515*	Xq21 deletion	2157_2158delTG,719fs*1	p.Ile22Thr
p.Ile244Thr	p.Arg743*			
	p.709 fs*1			
	p.Glu519*			
	p.Arg721*			
	p.Cys706Arg			

## Data Availability

The data supporting the conclusions of this article will be made available by the authors without undue reservation.

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
