# Peer review of "Electroencephalographic and Epilepsy Findings in ZNF711 Variants: A Case Series of Two Siblings"

_2035-8377, 2025, doi:10.3390/neurolint17010014_

Round 1
Reviewer 1 Report
Comments and Suggestions for Authors
This is a case-series consisting of two cases with phenotypic spectrum of ZNF711 gene. The major concern is poor and vague writing. It should be medically and scientifically revised. The main finding of study and its clinical application and relevance should be focused and discussed.
pathogenic variantsThe language of paper needs to be improved significantly. There are many grammatical and scientific mistakes throughout the paper.
The case presentations lack enough details on the patients medical history.
The relation between 2 cases has not been clearly mentioned in case presentation.
How the clinicians were suspicious about genetic disorders? How the other causes of epilepsy were excluded? There is not enough cohesion between different parts of the case presentation.
Comments on the Quality of English Languagevague writing and grammatical mistakes
Author Response
Comment 1: The major concern is poor and vague writing. It should be medically and scientifically revised. The main finding of study and its clinical application and relevance should be focused and discussed...The language of paper needs to be improved significantly. There are many grammatical and scientific mistakes throughout the paper.
Response 1: Thank you very much for the suggestions. We have edited the discussion to make it clearer and richer. We also tried to correct grammatical errors.We hope it is better now.
Comment 2: The case presentations lack enough details on the patients medical history. The relation between 2 cases has not been clearly mentioned in case presentation.
Response 2: We have enriched the medical history of the two brothers with more details, from line 77 to line 150. We have also better specified the relationship between the two brothers both from the beginning of the case report description to line 71. "We report a familiar case series of two male siblings presenting with mild to moderate ID, language delay, motor stereotypies and generalized seizure". Also specified in the title is the relationship of the two patients: Electroencephalographic and epilepsy findings in ZNF711 variants: a case series of two siblings.
Comment 3: How the clinicians were suspicious about genetic disorders? How the other causes of epilepsy were excluded? There is not enough cohesion between different parts of the case presentation.
Response 3: Causes of structural brain pathology were ruled out by performing MRI of the brain; then hereditary metabolic pathology by blood and urine laboratory tests; finally, genetic causes were evaluated. We have changed the approach of case presentation by making it more precise from line 77 to line 150.
Thank you very much for all the comments. I hope my answers have been comprehensive and the work will turn out richer.
Reviewer 2 Report
Comments and Suggestions for Authors
The title could be improved by mentioning that it is a case report.
A validated test is necessary to demonstrate the presence of intellectual disability. The apparent delay in achieving certain developmental milestones is insufficient for diagnosing intellectual disability, as these may fall within the spectrum of normal development, considering the intervals or windows for milestones. Additionally, the authors mention grading intellectual disability from mild to moderate, which is not substantiated.
The introduction requires a stronger theoretical framework to suggest a potential relationship between the c.657C>G variant, seizures, and intellectual disability.
The information in lines 28–31 lacks context regarding age or condition and appears to have a tenuous connection to the case presented.
The case presentation needs a chronological order of events and needs to explain the rationale behind specific tests. This information could be critical for a clinician interested in the case, as they might encounter similar conditions in their practice. For instance, in Proban 1, it is impossible to determine the subject’s age and, consequently, the timing of the tests performed. Seizures at the age of 3 are mentioned, followed by an MRI, without explaining the decision-making process or the patient’s age at the time of the MRI. A similar issue arises in the presentation of the second case, where the rationale for the applied treatment is not provided.
The use of abbreviations needs to be standardized. Some are explained more than once (e.g., lines 78 vs. 100), while others, such as “valproate,” are abbreviated but not subsequently used.
The figures are not mentioned in the text and lack essential details, such as the baseline conditions under which the measurements were taken.
The discussion does not address the proposed variant’s relevance, which is critical for the manuscript’s scientific integrity.
Lines 161–209 appear to be an oversight on the authors’ part.
There needs to be a mention of adherence to a case report presentation guideline (like CARE) or the ethical principles followed by the authors in handling, obtaining, and publishing patient data.
Author Response
Comment 1: The title could be improved by mentioning that it is a case report
Response 1: Thank you very much for the comment, I agree. We have modified the title as follows: "Electroencephalographic and epilepsy findings in ZNF711 variants: a case series of two siblings."
Comment 2: "A validated test is necessary to demonstrate the presence of intellectual disability. The apparent delay in achieving certain developmental milestones is insufficient for diagnosing intellectual disability, as these may fall within the spectrum of normal development, considering the intervals or windows for milestones. Additionally, the authors mention grading intellectual disability from mild to moderate, which is not substantiated."
Response 2: I agree with your comment, I apologize for the inaccuracy. We have added the cognitive test used. For patient 1: "Cognitive evaluation with Griffiths mental development scale performed at 3 years and 9 months of age, showed a moderate ID (GQ 56)" on lines 96-97; for patient 2: "At the age of 4 years, the patient has been tested with Griffiths mental development scale, with the result of a a moderate ID (GQ 47)." on lines 128-129.
Comment 3: The introduction requires a stronger theoretical framework to suggest a potential relationship between the c.657C>G variant, seizures, and intellectual disability.
Response 3: We tried to explain this relationship by modifying the introduction from line 50 like this: "According to the evidence currently present in literature, ZNF711 seems to be associated with a non-syndromic mild-to-moderate intellectual disability which is associated with occasional comorbidities, such as autistic features and speech delay. Hence, we describe the clinical and EEG features of two siblings carrying a hemizygous c.657C>G, pathogenic variant in the ZNF711 gene, affected by a non syndromic form of ID and seizures.It was hypotized a role for zing finger proteins (ZNFs) in neurological development: particularly, numerous studies have shown that ZNFs may regulate gene expression at the transcriptional and translational levels and seem to be involved in the sonic hedgehog signalling pathway, which has been directly associated with seizure genesis. This evidence supported the idea of a clinical relevance of the ZNF711 variant encountered in our patients, with both ID and seizures."
Comment 4: The information in lines 28–31 lacks context regarding age or condition and appears to have a tenuous connection to the case presented.
Response 4: we added information regarding the age and the funzionale status of the patients, specifically for patient 1 we inserted this sentence in line 98: “At the last clinical evaluation (16 years old), he showed persistence of motor stereotypies, regression of behavioral problems and seizure-free outcomes since VPA was started; for patient 2: “At the last clinical evaluation at 12 years of age, the patient presented persistence of motor stereotypies, expressive language characterized mainly by bisyllabic words, supported by communication through pointing, and seizure-free outcomes since four years of age with VPA monotherapy” in line 130.
Comment 5: The case presentation needs a chronological order of events and needs to explain the rationale behind specific tests. This information could be critical for a clinician interested in the case, as they might encounter similar conditions in their practice. For instance, in Proban 1, it is impossible to determine the subject’s age and, consequently, the timing of the tests performed. Seizures at the age of 3 are mentioned, followed by an MRI, without explaining the decision-making process or the patient’s age at the time of the MRI. A similar issue arises in the presentation of the second case, where the rationale for the applied treatment is not provided.
Response 5: Thank you for this suggestion. We have tried to give a more precise chronological order to the events, from line 76 to line 150. I hope it is clearer.
Comment 6: The use of abbreviations needs to be standardized. Some are explained more than once (e.g., lines 78 vs. 100), while others, such as “valproate,” are abbreviated but not subsequently used.
Response 6: Of course, I agree. I think I changed all the abbreviations in the text
Comment 7: The figures are not mentioned in the text and lack essential details, such as the baseline conditions under which the measurements were taken.
Response 7: we modified the text as follows: "Such epileptic manifestations are depicted in figure 1: they are characterized by jerks in the upper limbs with an ictal EEG showing generalized slow spike-waves and EMG signals congruent with myoclonic seizures. (figure 1)." on line 109; for patient 2: "Interictal activity showed slow spike-waves in the right posterior regions, with additional spike-waves and polyspike-waves in the anterior regions (figure 2)." on line 136.
Comment 8: The discussion does not address the proposed variant’s relevance, which is critical for the manuscript’s scientific integrity.
Response 8: Thank you for the valuable advice. We have edited the discussion to show the relevance of the variant. In particular, we pointed out that there are no other described cases in which seizures are present and that we think it plausible that this variant may play a role in the development of epilepsy also in light of studies in the literature on the role of zinc finger proteins in the development of neurological disorders, including epilepsy. Particularly, within the article we added: "There is paucity of information regarding the physiopathological role of zinc finger proteins (ZNF) in seizures. However, several studies demonstrated that these genes may have a role in transcriptional and translational regulation of other genes involved in central nervous system development through the sonic hedgehog (SHH) signaling pathway. Indeed, some proteins belonging to the same family of ZNF711 have been proved to have a relationship not only with neuronal inflammation and oxidative stress seen in stroke and other diseases but also in stem cells proliferation and neuronal differentiation associated with autism and schizophrenia [5]It has been widely demonstrated that SHH is involved in epilepsy through an enhancement of extracellular glutamate levels which trigger neuronal excitation leading to epilepsy [13]. Being ZNF important modulators of SHH signaling pathways, it can hypotized a role of these genes in neurological development and epilepsy [12]"
Comment 9: Lines 161–209 appear to be an oversight on the authors’ part
Response 9: thank you, I deleted them
Reviewer 3 Report
Comments and Suggestions for Authors
Dear Author;
Below are your comments on your article;
‘Electroencephalographic and epilepsy findings in ZNF711 variants: A family study’ changing the punctuation marks in the title will make it more understandable.
The normal timing of gait is accepted as 12-18 months; if there is no independent walking after the 18th month, it is considered delayed. However, if there is a delay in motor milestones such as head control and sitting without support according to chronological age, you can specify these.
You can use ‘case’ instead of ‘proband’.
‘intellectual disability’ should be abbreviated in its first use and continued with its abbreviated form in subsequent uses.
Is there any information about the two siblings’ current age and functional status?
References should be described as follows:
- Journal Articles:
1. Author 1, A.B.; Author 2, C.D. Title of the article. Abbreviated Journal Name Year, Volume, page range.
Author Response
Comment 1: ‘Electroencephalographic and epilepsy findings in ZNF711 variants: A family study’ changing the punctuation marks in the title will make it more understandable.
Answer 1: Many thanks for your comments. We changed the title as follows: "Electroencephalographic and epilepsy findings in ZNF711 variants: a case series of two siblings."
Comment 2: 'The normal timing of gait is accepted as 12-18 months; if there is no independent walking after the 18th month, it is considered delayed. However, if there is a delay in motor milestones such as head control and sitting without support according to chronological age, you can specify these.'
Answer 2: I apologize for the inaccuracy, we have better specified the acquisition of the neuromotor developmental stages of the two siblings. Specifically for patient 1, we modified on line 80 as follows: "His psychomotor developmental milestones were characterized by independent walking at 15 months and mild language delay, with initial words at around 2 years of age." Whereas for patient 2: "His psychomotor developmental milestones were characterized by independent walking at 18 months and language delay, with first words after the age of two and scarce progressive lexical acquisition" on line 119.
Comment 3: You can use ‘case’ instead of ‘proband’.
Answer 3: we replaced "case" instead of "proband", as suggested (lines 76 and 115)
Comment 4: ‘intellectual disability’ should be abbreviated
Answer 4: we abbreviated 'intellectual disability' to 'ID'
Comment 5: Is there any information about the two siblings’ current age and functional status?
Answer 5: We have added the current clinical information of the two siblings as follows. For patient 1 we inserted this sentence in line 98: “At the last clinical evaluation (16 years old), he showed persistence of motor stereotypies, regression of behavioral problems and seizure-free outcomes since VPA was started; for patient 2: “At the last clinical evaluation at 12 years of age, the patient presented persistence of motor stereotypies, expressive language characterized mainly by bisyllabic words, supported by communication through pointing, and seizure-free outcomes since four years of age with VPA monotherapy” in line 130.
Comment 6: 'References should be described as follows...'
Answer 6: We changed the references as follows: for example: '1. K.D.; C.J. A functional link between the histone demethylase PHF8 and the transcription factor ZNF711 in X-linked mental retardation. Mol Cell. 2010' 'on line 255
Thank you again for the comments provided, I hope my answers were comprehensive.
Reviewer 4 Report
Comments and Suggestions for Authors
The authors presented the case of two male siblings of intellectual disability associated with seizures in probands carrying a hemizygous c.657C>G pathogenic variant in the ZNF711 gene. The case study intends to expand the clinical spectrum of ZNF711 variants by focusing on epilepsy as a potential comorbidity.
The authors submitted a well written manuscript. I have a few suggestions for a revised version.
Major:
1. Are there any images/figures to show the sequencing/mapping, and identification of the hemizygous c.657C>G, pathogenic variant in the ZNF711 gene of the males reported in the study? If so, include the images/references.
2. Include a table with the literature referenced in the manuscript with the symptoms, genes involved, references and any other relevant information.
3. Is there any information related to any longitudinal studies or latest information on these subjects?
Minor:
Lines 127-128 are kind of repetitive to the lines in 105-106. May be combine both if you want to provide additional details.
Author Response
Comment 1: Are there any images/figures to show the sequencing/mapping, and identification of the hemizygous c.657C>G, pathogenic variant in the ZNF711 gene of the males reported in the study? If so, include the images/references.
Answer 1: Thank you very much for the suggestions. I am very sorry, but unfortunately there are no images depicting the sequencing of the ZNF711 variant of the two sibling
Comment 2: Include a table with the literature referenced in the manuscript with the symptoms, genes involved, references and any other relevant information.
Answer 2: Thank you for this suggestion, you can find the table attached
Comment 3: Is there any information related to any longitudinal studies or latest information on these subjects?
Answer 3: We have added the most recent information on the study subjects; specifically for patient 1 we inserted this sentence in line 98: “At the last clinical evaluation (16 years old), he showed persistence of motor stereotypies, regression of behavioral problems and seizure-free outcomes since VPA was started; for patient 2: “At the last clinical evaluation at 12 years of age, the patient presented persistence of motor stereotypies, expressive language characterized mainly by bisyllabic words, supported by communication through pointing, and seizure-free outcomes since four years of age with VPA monotherapy” in line 130.
Reviewer 5 Report
Comments and Suggestions for Authors
The authors presented an interesting case of two boys with intellectual disability (ID) and developmental disorders who experienced generalized tonic-clonic seizures. The boys were subject of genetic testing which revealed a hemizygous c.657C>G, pathogenic variant in the ZNF711 gene. The value of the conducted research for science is that it documents the possibility of epileptic seizures in people with ID who are carriers of a hemizygous c.657C>G pathogenic variant in the ZNF711 gene. This result can be used in diagnostic practice. However, in the text of the article, I noticed some gaps that require supplementation. For example, in the description of the boys studied, there is no information about their age, the health situation of their parents and the method of qualification for the study. After supplementing this information, I support the publication of the reviewed article.
Author Response
Comment: "there is no information about their age, the health situation of their parents and the method of qualification for the study"
Answer: Many thanks for your precious comment, we added information regarding the age of the patients, specifically for patient 1 we inserted this sentence in line 98: “At the last clinical evaluation (16 years old), he showed persistence of motor stereotypies, regression of behavioral problems and seizure-free outcomes since VPA was started; for patient 2: “At the last clinical evaluation at 12 years of age, the patient presented persistence of motor stereotypies, expressive language characterized mainly by bisyllabic words, supported by communication through pointing, and seizure-free outcomes since four years of age with VPA monotherapy” in line 130.
Whereas regarding parental health status: “No previous history of epilepsy or neurodevelopmental delays was mentioned in the family and parents” in line 74.
Furthermore, regarding the qualification methods of the study, we specified in the introduction that since the case is very particular and rare, we decided to study the case history present in the literature with this case series: "Considered the paucity of information regarding this rare genetic condition, our aim with this familiar case series is to extend the clinical features of the phenotypic spectrum non- syndromic ZNF711-related ID." in line 66.
Again many thanks for the comments
Round 2
Reviewer 1 Report
Comments and Suggestions for Authors
My comments have been addressed.
thanks
Author Response
thank you very much again for your comments, best regards
Reviewer 2 Report
Comments and Suggestions for Authors
The authors should ensure uniformity in their writing style, text formatting, and adherence to a reference style that meets a basic standard.
1. Line 33: Ensure the correct punctuation preceding and following the abbreviation “etc.,” which should include a comma and a period.
2. Line 36: The acronym XLID appears misplaced and unclear, as its meaning or context is not defined. Please clarify or provide its definition at first mention.
3. Line 47: The authors refer to families carrying this mutation, which has been previously described. It is recommended that appropriate references be included to support this statement.
4. Line 54: Verify that the punctuation used in the citation context complies with the journal’s style and grammar rules. In this case, the reference appears between colons, as if it were a complete sentence. Similar punctuation errors around citations are observed in lines 113, 183, 199, 201, and 235 (e.g., Liang et al., 2017).
5. Lines 58–61: Consider moving this information to the paragraph discussing zinc finger proteins (lines 47–51) to strengthen the connection between the theory and the relevance of this gene. Additionally, reorganize the introduction, as the section from lines 63–70 includes three one-sentence paragraphs presenting isolated ideas.
6. Lines 55, 98, 104, 112, 145: Abbreviations are undefined or defined after their first appearance. Review each abbreviation and ensure proper definition upon its first use unless it is an official symbol.
7. Lines 81, 119: The description of the physical evaluation in the newborn mentions indicators such as small and spaced teeth, which are not typically observable at this stage. Could the authors clarify how or in what context this evaluation was conducted?
8. Lines 93–102: These lines contain five one-sentence paragraphs. It is recommended that the authors provide more continuity and more precise development of the case description. For example, line 95 repeats the child’s age (3 years) mentioned in line 87, creating unnecessary redundancy.
9. Line 99: To highlight the clinical case’s significance, could the authors clarify why this clinical evaluation was conducted?
10. Line 99: The authors use “stereotype” instead of “stereotypy,” which persists as an error in subsequent mentions.
11. Lines 110, 129: It is suggested to use the past perfect tense when describing the case.
12. Line 142: Could the authors provide the context that led to the decision to perform an EEG?
13. Line 146: Similarly, to emphasize the significance of the clinical case, it would be valuable for the reader to understand the context in which a genetic sequencing procedure was performed. Did the combination of signs and symptoms prompt this approach? Was it part of a specific research protocol offered to the patients? Or was this pursued by the parents outside of a clinical framework?
14. Line 156: The figure states the EEG was performed at 5 years of age, but the text indicates it was done at 4. Which is correct?
15. Lines 166–167: The authors present a classical definition of intellectual disability, but its relevance at this point in the discussion is unclear. Could this be expanded or better integrated into the argument?
16. Table 1: Regarding the ZNF711 variant, can the authors confirm that it is the same variant under study? If so, please explicitly state this in the table.
17. Line 195: The term “mental retardation” is considered outdated and potentially offensive. Please consider using “intellectual disability” or a more current term.
18. Line 227: The authors mention “several studies” to support the role of the gene being studied. It is recommended that specific references be included to substantiate this claim.
19. Conclusions: The authors should consider including the limitations of their case report in the conclusions to provide proper context for their findings.
Author Response
Comment 1: The authors should ensure uniformity in their writing style, text formatting, and adherence to a reference style that meets a basic standard
Response 1: Thank you, I have modified style and text formatting
Comment 2: Line 33: Ensure the correct punctuation preceding and following the abbreviation “etc.,” which should include a comma and a period.
Response 2: Thank you for your comment, I corrected the abbreviation etc.
Comment 3: Line 36: The acronym XLID appears misplaced and unclear, as its meaning or context is not defined. Please clarify or provide its definition at first mention.
Response 3: Thank you, I agree. I added this farse to line 36: "Overall, genes involved in ID are frequently located on the X chromosome and thus responsible for X-linked intellectual disability (XLID) which clearly affects male more than females."
Comment 4: Line 47: The authors refer to families carrying this mutation, which has been previously described. It is recommended that appropriate references be included to support this statement.
Response 4: Thank you, I have added the references
Comment 5: Line 54: Verify that the punctuation used in the citation context complies with the journal’s style and grammar rules. In this case, the reference appears between colons, as if it were a complete sentence. Similar punctuation errors around citations are observed in lines 113, 183, 199, 201, and 235 (e.g., Liang et al., 2017).
Response 5: thank you, I corrected the punctuation regarding the citations.
Comment 6: Lines 58–61: Consider moving this information to the paragraph discussing zinc finger proteins (lines 47–51) to strengthen the connection between the theory and the relevance of this gene. Additionally, reorganize the introduction, as the section from lines 63–70 includes three one-sentence paragraphs presenting isolated ideas.
Response 6: thank you for the suggestion, we reorganized the paragraph and added connections among the different sentences.
Comment 7: Lines 55, 98, 104, 112, 145: Abbreviations are undefined or defined after their first appearance. Review each abbreviation and ensure proper definition upon its first use unless it is an official symbol.
Response 7: thanks for the advice, I corrected the abbreviations
Comment 8: Lines 81, 119: The description of the physical evaluation in the newborn mentions indicators such as small and spaced teeth, which are not typically observable at this stage. Could the authors clarify how or in what context this evaluation was conducted?
Response 8: Thank you for your comment, I specified that the physical assessment was done at age 2 for both of them. For patient 1, I added this sentence: " On physical examination, at age of 2, he showed facial features included macrocephaly, sunken eyes, a flat philtrum, small and spaced teeth and flat feet" on line 92; for patient 2: "On that occasion, the objective examination showed adequate auxological parameters and - as his sibling - facial features included sunken eyes, flat philtrum, small/spaced teeth and flat feet." on line 130.
Comment 9: These lines contain five one-sentence paragraphs. It is recommended that the authors provide more continuity and more precise development of the case description. For example, line 95 repeats the child’s age (3 years) mentioned in line 87, creating unnecessary redundancy.
Response 9: Thank you for your comment, I have corrected the writing of the case report, removing redundant sentences
Comment 10: Line 99: To highlight the clinical case’s significance, could the authors clarify why this clinical evaluation was conducted?
Response 10: Thank you, we clarified that he was evaluated during periodical follow up in our Neurological Outpatient Clinic.
Comment 11: Line 99: The authors use “stereotype” instead of “stereotypy,” which persists as an error in subsequent mentions.
Response 11: thank you, I corrected the word "stereotypy"
Comment 12: Lines 110, 129: It is suggested to use the past perfect tense when describing the case.
Response 12: thank you, I corrected the use of the past perfect
Comment 13: Line 142: Could the authors provide the context that led to the decision to perform an EEG?
Response 13: Thank you I clarified in the text why we performed an EEG, that is in the suspicion of seizures
Comment 14: Line 146: Similarly, to emphasize the significance of the clinical case, it would be valuable for the reader to understand the context in which a genetic sequencing procedure was performed. Did the combination of signs and symptoms prompt this approach? Was it part of a specific research protocol offered to the patients? Or was this pursued by the parents outside of a clinical framework?
Response 14: thanks for the suggestion, I clarified why a genetic evaluation was performed, particularly for the combination of epilepsy and ID
Comment 15: Line 156: The figure states the EEG was performed at 5 years of age, but the text indicates it was done at 4. Which is correct?
Response 15: thank you. The correct age is 4 years, I corrected it
Comment 16: Lines 166–167: The authors present a classical definition of intellectual disability, but its relevance at this point in the discussion is unclear. Could this be expanded or better integrated into the argument?
Response 16: Thank you, I agree. I preferred to delete this sentence precisely because of its irrelevance at this point in the paper.
Comment 17: Regarding the ZNF711 variant, can the authors confirm that it is the same variant under study? If so, please explicitly state this in the table.
Response 17: The variant involves the ZNF711 gene, but it is not the same in all the studies in the table
Comment 18: Line 195: The term “mental retardation” is considered outdated and potentially offensive. Please consider using “intellectual disability” or a more current term.
Response 18: Thank you, I agree. I corrected it.
Comment 19: Line 227: The authors mention “several studies” to support the role of the gene being studied. It is recommended that specific references be included to substantiate this claim.
Response 19: Thank you, I added the references
Comment 20: Conclusions: The authors should consider including the limitations of their case report in the conclusions to provide proper context for their findings.
Response 20: Thank you, I have modified the conclusions as follows: "...This case is limited by the anecdotal nature of the report. While additional studies are required to better understand the specific role of ZNF711 variants in the development of epilepsy, it expands the clinical spectrum of ZNF711 variants by highlighting epilepsy as a possible comorbidity..."
Round 3
Reviewer 2 Report
Comments and Suggestions for Authors
I appreciate the authors’ efforts addressing the previous observations regarding their valuable work. To further emphasize the significance of the reported variant and its clinical association, I believe the following suggestions could enhance the impact of their study:
1. Line 47: It may be beneficial to include an approximate or updated number of ZNF711 variants associated with intellectual disability (ID).
2. Based on the authors’ response regarding the term “ZNF711 variants” in Table 1, which they indicate could refer to any variant, mentioning the specific variant studied in the cases discussed (Van der Werf, line 199; Wang, line 203; Tarpey, line 217) could help underscore the relevance of the variant reported in the present manuscript.
Author Response
Comment 1: It may be beneficial to include an approximate or updated number of ZNF711 variants associated with intellectual disability (ID).
Response 1: thanks for the comment. I have added this sentence in the discussion in this regard: "To the best of our knowledge, by conducting a narrative review of the literature, about 10 variants of ZNF711 have been reported to be associated with ID (table 2)."on line 183
Comment 2: Based on the authors’ response regarding the term “ZNF711 variants” in Table 1, which they indicate could refer to any variant, mentioning the specific variant studied in the cases discussed (Van der Werf, line 199; Wang, line 203; Tarpey, line 217) could help underscore the relevance of the variant reported in the present manuscript.
Response 2: I have added a table (table 2) with the described variants of ZNF711.
Thanks for the suggestions.